# Controlled-Release Nitrogen Mixed with Common Nitrogen Fertilizer Can Maintain High Yield of Rapeseed and Improve Nitrogen Utilization Efficiency

**DOI:** 10.3390/plants12244105

**Published:** 2023-12-08

**Authors:** Yue Hu, Fangfang Zhang, Hafiz Hassan Javed, Xiao Peng, Honglin Chen, Weiqun Tang, Ying Lai, Yongcheng Wu

**Affiliations:** 1College of Agronomy, Sichuan Agricultural University, Chengdu 611130, China; 13667654705@163.com (Y.H.); 13915156588@163.com (F.Z.); hafizhassanjaved@yahoo.com (H.H.J.); 13858579931@163.com (X.P.); 13991952258@163.com (W.T.); 13888587228@163.com (Y.L.); 2Key Laboratory of Crop Ecophysiology and Farming System in Southwest China, Chengdu 611130, China; 3Sichuan Academy of Agricultural Sciences, Chengdu 610066, China; 13558895655@163.com

**Keywords:** reduce nitrogen fertilizer, microbial biomass carbon and nitrogen, soil enzyme activity, ammonium nitrogen, nitrate nitrogen

## Abstract

Field experiments were conducted to study the effects of different proportions of controlled-release nitrogen fertilizer mixed with quick-acting nitrogen fertilizer on the yield and nitrogen utilization efficiency of direct-seeding rapeseed. Using a conventional nitrogen application rate of 180 kg ha^−1^ as a control, a total of 5 types of available nitrogen fertilizers and different proportions of controlled-release nitrogen fertilizers were mixed for fertilizer treatment. The proportion of available nitrogen fertilizer used was 135 kg ha^−1^, and the addition ratios of the five types of controlled-release nitrogen fertilizers were 0%, 30%, 50%, 70%, and 100%, respectively (i.e., the proportion of controlled-release nitrogen to the total nitrogen application amount). These ratios were represented as N135R0, N135R1, N135R2, N135R3, and N135R4, respectively. The results showed that there was no significant difference in the number of pods per plant, the number of seeds per pod, or the grain yield under the treatment of controlled-release nitrogen fertilizer mixed with quick-acting nitrogen fertilizer for proportions of 30–50% (N135R1~R3) when compared with the control, and a stable yield was achieved. Mixing controlled-release nitrogen fertilizer under reduced nitrogen application can significantly improve the apparent utilization rate of rapeseed nitrogen fertilizer, but it first increases and then decreases with the increase of the controlled-release nitrogen mixing ratio, reaching its highest under the N135R2 treatment. The agronomic utilization efficiency and partial productivity of nitrogen fertilizer first increased and then decreased with the increased proportion of controlled-release nitrogen, and both reached their highest utilization with the N135R2 treatment. The mixed treatment of controlled-release nitrogen did not affect soil urease activity, but significantly increased soil sucrase activity. The mixed treatment of controlled-release nitrogen also increased soil microbial biomass nitrogen and carbon content. Especially in the flowering stage, the soil microbial biomass nitrogen and carbon content was significantly higher under a controlled-release nitrogen mixing ratio of 30–50%. At the same time, it had a similar effect on soil inorganic nitrogen content. Therefore, a controlled-release nitrogen mixing treatment provided sufficient nitrogen for the key growth period of rapeseed. Under the condition of reducing the amount of nitrogen fertilizer by 25% based on the amount of nitrogen fertilizer applied to conventional rapeseed, the application of controlled-release urea mixed with common nitrogen fertilizer mixed at a ratio of 30–50% can be an effective way to maintain grain yield levels and improve nitrogen utilization efficiency.

## 1. Introduction

Rapeseed is one of the world’s major oil crops, and China’s rapeseed planting area and production rank among the world’s top [1]. Rapeseed is a major winter crop grown in the Yangtze river basin in China and plays an important role in the rural economy [2]. Rapeseed is a crop that requires a lot of fertilizer, and its yield largely depends on the application of exogenous fertilizers, especially nitrogen. Nitrogen is one of the most important nutrients in the growth and development of rapeseed [3]. The proper application of nitrogen fertilizer can promote nitrogen uptake and distribution in rapeseed plants and improve photosynthesis in the plant, ultimately leading to increased yields [4]. People use a lot of nitrogen fertilizer to improve crop yields, leading to a nationwide increase in the application of nitrogen fertilizer year over year in recent years. Still, nitrogen utilization efficiency in China is lower than that of developed countries [5,6]. Studies have shown that winter rapeseed has a long growth cycle and that splitting fertilizer applications can improve yield and fertilizer utilization [7]. However, with the increasing urbanization of China leading to an exodus and reduction of rural labor, labor costs are increasing; additionally, the split application of nitrogen fertilizer is time-consuming and labor-intensive, hence the use of one-off fertilizer applications is more common [8]. However, although a single application of common nitrogen fertilizer can reduce labor costs and meet the nutrient requirements of rapeseed in the early stages of growth, a lack of nutrient supply in the later stages of growth leads to a reduction in rapeseed yield and also poses a risk of environmental pollution and reduced resource use efficiency [9,10].

Compared to common urea, slow-release nitrogen fertilizer is a new type of fertilizer that is recommended for long term use as a base fertilizer to reduce production costs and increase crop yield by reducing nitrogen losses due to the slow release of nutrients [11,12]. Research shows that the reasonable application of controlled-release N fertilizer can reduce losses such as gassing and leaching of N fertilizer, improve N fertilizer utilization and increase yield, and save 30–50% of N fertilizer [13,14]. However, the release of nutrients from controlled-release nitrogen fertilizers is strongly influenced by environmental factors. The application of controlled-release N fertilizer alone may have adverse effects on the crop or the environment when the weather is harsh; for example, low temperatures or drought in the early stages of fertility may slow down the release of nutrients and affect crop growth, leaving too much inorganic nitrogen in the soil and increasing ecological risks [15]. On the other hand, the relatively high price of controlled-release nitrogen fertilizer makes it difficult for farmers to use on a large scale [16]. Some researchers have therefore proposed mixing controlled-release nitrogen fertilizers with common urea in appropriate proportions, and to regulate the supply of fast-acting and slow-acting nitrogen to meet the nitrogen requirements of crops at different times, while also saving costs and increasing efficiency [17]. Therefore, we assume that a mixed application of controlled-release nitrogen fertilizer and conventional urea can maintain high yield and improve nitrogen utilization efficiency in rapeseed. The objectives of this study were (1) to evaluate the effect of a mixed application of controlled-release nitrogen fertilizer and conventional urea on soil carbon and nitrogen nutrition, (2) to determine the effect of a mixed application of controlled-release nitrogen fertilizer and conventional urea on yield components, and (3) to clarify the effect of a mixed application of controlled-release nitrogen fertilizer and conventional urea on improving nitrogen fertilizer utilization efficiency. To verify the hypothesis, we conducted a two-year field experiment. A conventional N use of 180 kg N ha^−1^ was used as control, and a 135 kg N ha^−1^ reduced N application for different controlled-release N fertilizer blending ratios (0, 30%, 50%, 70%, 100%) was set up on this basis, expressed as R0, R1, R2, R3, and R4, respectively. This study investigates the reasonable blending ratio of controlled-release N fertilizer under reduced N applications in a field of rice stubble direct-seeded with rapeseed, and the results provide scientific support for reduced and efficient N fertilizer management in direct-seeded rapeseed crops.

## 2. Materials and Methods

### 2.1. Experimental Materials

The rapeseed variety Mianbangyou was chosen for the study, a variety with a vast cultivation area in the Yangtze river’s upper reaches. The controlled-release nitrogen fertilizer used in the study is resin-coated urea, provided by China Lvzhou Fertilizer Technology Co., Ltd. (Fujian, China), with a nitrogen content of 44%. This controlled-release nitrogen fertilizer has the characteristics of slow-release speed and good persistence, making it suitable for crops with long growth cycles. Conventional urea with a nitrogen content of 46% was purchased from the local agricultural material market. The nutrient contents of the phosphate fertilizer and potassium fertilizer were 12% P_2_O_5_ and 60% K_2_O, respectively.

### 2.2. Experimental Site

The experiment was conducted at the rapeseed oil production base in Xigao Town, Guanghan City, Sichuan Province, from October 2018 to May 2020. The region has a humid central subtropical climate zone with an average annual temperature of 16.3 °C, average annual precipitation of 890.8 mm, and average annual sunshine of 1229.2 h. The key meteorological data collected over two years are displayed in Figure 1. The stubble before the experiment was conducted was rice, and the soil was paddy soil. Soil samples were taken and analyzed from 0 to 20 cm of the soil tillage layer before the rapeseed was sown. The basic soil fertility consisted of 30.87 g/kg organic matter, 1.83 g/kg total nitrogen, 30.93 mg/kg available phosphorus, 96 mg/kg available potassium, 27.26 mg/kg available nitrogen, and a 6.19 pH value.

### 2.3. Experiment Design and Field Management

The experiment was designed as a one-way randomized group trial. Five treatments of controlled-release N fertilizer blended with fast-acting N fertilizer (controlled-release N as a percentage of total N applied) were set up based on a reduced N application of 135 kg ha^−1^ (N135), each at 0%, 30%, 50%, 70%, and 100% (denoted as N135R0, N135R1, N135R2, N135R3 and N135R4, respectively). A conventional N application of 180 kg ha^−1^ (N180 CK, common N fertilizer) was used as a control. The experiment consisted of 6 treatments, each replicated 3 times, with a total of 18 plots. The experimental plot size was 20 m^2^ (4 m × 5 m). The trial was sown in early October each year using hand-drawn lines at a spacing of 33 cm between rows and 20 cm between holes, leaving 2 plants in each hole, and the seedlings were set at a density of 300,000 plants ha^−1^ at the 3–5 leaf stage. Nitrogen fertilizer was applied in a single application as a low fertilizer, and phosphorus, potassium, and boron fertilizer at a rate of 60 kg ha^−1^ P_2_O_5_, 90 kg ha^−1^ K_2_O, and 15 kg ha^−1^ borax were applied in a single application as a base fertilizer. All fertilizer applications were made via hand spreading and mixing the cultivated soil with a hoe. One month after emergence, unified herbicides and insecticides were used to prevent and control pests and diseases.

### 2.4. Sampling and Measurements

#### 2.4.1. Soil Sampling and Determination

At the seedling and flowering stages of each growing season, 5-point samples were taken from each plot at 0–20 cm to make a composite sample and then brought back to the laboratory for testing. Refer to the determination method of Li et al. for the content of nitrate nitrogen and ammonium nitrogen in soil [18]. Soil microbial biomass carbon and nitrogen were determined via fumigation method [19]. Soil urease activity was measured using the phenol sodium hypochlorite colorimetric method, while sucrase activity was determined by using the 5-dinitrosalicylic acid colorimetric method [20].

#### 2.4.2. Yield and Yield Components

At the mature stage of the rapeseed plant, 10 representative plants were selected from each plot to determine yield components; effective pod number per plant, seed number per pod, and 1000-grain weight were investigated. Each plot was harvested manually, and the yield was measured after drying for one week.

#### 2.4.3. Nitrogen Content

After ripening and harvesting, the plant samples were divided into stems and pods and seeds, and then dried to a constant weight at a constant 80 °C oven temperature. The total nitrogen content of each plant organ was measured using an automatic Kjeldahl nitrogen analyzer (FOSS 8400, FOSS Group Corporation, Beijing, China).

#### 2.4.4. Calculation Methods for Parameters Related to Nitrogen Use

Parameters related to nitrogen use were calculated using the following formulas [21]:Agronomic efficiency of N (AEN, kg kg^−1^) = (Crop yield in N area − Crop yield in non-N area)/amount of N applied(1)
Partial factor productivity of N (PFPN, kg kg^−1^) = Seed yield in N application area/N application(2)
Nitrogen harvest index (NHI, %) = Total grain nitrogen accumulation/Total plant nitrogen accumulation(3)
Apparent recovery efficiency of N (AREN, %) = (N uptake by crop in N-fertilized area − N uptake by crop in non-fertilized area) × 100/N rate.(4)

### 2.5. Statistics Analysis

Microsoft Excel 2019 was used for data arrangement. Analysis of variance was performed using SPSS Software 19.0 followed by the least significant difference (LSD) test to assess the differences among treatments at a probability level of 0.05.

## 3. Results

### 3.1. Effects of Different Fertilization Treatments on Soil Microbial Carbon and Nitrogen Content and Enzyme Activity

At the seedling stage, soil microbial biomass carbon and microbial biomass nitrogen content increased with the increased mixing proportion of controlled-release nitrogen (Figure 2). The maximum values over two years appeared in the N135R4 and N135R3 treatments, respectively. Compared with the CK, soil microbial biomass carbon increased by 4.8 and 5.3%, and soil microbial biomass nitrogen increased by 26.3 and 15.8%, respectively. Controlled-release nitrogen had no significant effect on urease activity but had a significant effect on invertase activity. The invertase activity increased with the proportion of controlled-release nitrogen over the two years. The maximum values were found in the N135R4 and N135R3 treatments, with 53.2 and 30.8% increases, respectively, compared to the CK. The trends of these four indicators after entering the flowering stage were generally consistent with those of the seedling stage. Soil microbial carbon and nitrogen content and urease activity increased compared to the seedling stage, while invertase activity showed a decrease (Figure 3).

### 3.2. Effect of Different Fertilizer Treatments on Soil Inorganic N Content

Over the two years, the content of soil ammonium nitrogen under the N135 treatment at the seedling stage first increased and then decreased with an increase in the proportion of controlled-release nitrogen (Figure 4A). The maximum values were found in the N135R2 and N135R3 treatments, with 70.7 and 67.3% increases, respectively, compared to the CK. This trend did not change overall by the flowering stage (Figure 4B), but ammonium N content decreased compared to the seedling stage under all treatments, and the maximum values were found in the N135R4 treatment. Compared with the CK, it increased by 14.1% and 6.6%, respectively, in two years. The changing trend of soil nitrate nitrogen content is similar to that of ammonium nitrogen. There was no significant difference in nitrate nitrogen content among different treatments during the seedling stage (Figure 4C). However, the nitrate nitrogen content under each treatment during the flowering period was significantly lower than the control, showing a similar trend of change over the two years (Figure 4D).

### 3.3. Yield and Yield Components

Under reduced nitrogen application (N135R0), the number of pods, seed yield, and biological yield of rapeseed per plant tended to increase and decrease as the controlled-release N fertilizer blending ratio increased. The maximum seed yield in both years occurred under the N135R2 treatment (Table 1) but was not significantly different compared to the CK. In 2018–2019, seed yield and yield components parameters were not significantly different under treatments N135R1 to N135R3 when compared to the CK. Biomass under treatment N135R2 was 5.2% higher than the control. In 2019–2020, the number of corms per plant, 1000-grain weight, and seed yield were not significantly different under the N135R2 treatment compared to the CK, while the number of seeds per pod and biological yield increased by 9.2 and 10.2%, respectively, compared to the CK. The combined results of the two years showed that treatments with a 33–67% controlled-release N blending under reduced N application could achieve reduced N and stable yields compared to the CK.

### 3.4. Nitrogen Uptake and Utilization

Different mixing proportions of controlled-release nitrogen can affect nitrogen partial factor productivity, agronomic utilization rates, and apparent nitrogen utilization rates of direct-seeding rapeseed (Table 2). From 2018 to 2019, with an increase of the mixing proportion of controlled-release nitrogen fertilizer, the agronomic utilization rate of nitrogen fertilizer and the partial productivity of nitrogen fertilizer first increased and then decreased. Under the reduced nitrogen application, the partial productivity, agronomic utilization rate, and apparent utilization rate of nitrogen fertilizer for the controlled-release nitrogen fertilizer mixing proportion treatments (N135R1, N135R2, N135R3, and N135R4) were significantly different from the CK. The trend of agronomic utilization rate, partial productivity, and apparent utilization rate of nitrogen fertilizer in 2019–2020 was the same as in 2018–2019. Two years of experiments showed that the mixing ratio of different controlled-release fertilizers had no significant effect on the nitrogen harvest index.

## 4. Discussion

Adequate nitrogen supply can promote the absorption of nitrogen by rapeseed plants, increase dry matter accumulation, and ultimately increase yield [22]. In this study, the yield of rapeseed was significantly reduced under reduced N treatment (N135), while blending controlled-release N fertilizer under reduced N conditions could maintain the yield of rapeseed. This may be due to the slow nutrient release of controlled-release nitrogen fertilizer, which makes rapeseed well synchronized with nitrogen supply during the whole growth period [23]. Mi et al. [24] reported that about 80% of the N from controlled-release urea was released before the heading stage in late and single rice; therefore, nitrogen supply is excessive in the early stage and insufficient in the later stage. In addition, controlled-release nitrogen fertilizer will increase root activity, increase nitrogen uptake, and thus promote plant growth [25]. In the nitrogen-reduction treatment (N135), the yield of the treatment with a controlled-release fertilizer blending ratio of 50–70% was higher than that of the CK. Meanwhile, the N135R2 treatment not only did not reduce yield, but actually increased yield by 1.3% compared with the CK. This indicates that applying a suitable controlled-release blending ratio can achieve the effect of nitrogen reduction and yield increase in rapeseed production, in which the best yield-increase effect is achieved by blending controlled-release at 50%.

Some studies have reported that the physiological factors for controlled-release to improve nitrogen use efficiency include biomass, nitrogen accumulation, and photosynthetic rate [26]. A high controlled-release N fertilizer blending ratio treatment has higher inorganic N content than a low controlled-release N fertilizer blending ratio treatment, thus ensuring the uptake and utilization of nitrogen by rapeseed in the middle and late stages [14]. In this study, plant nitrogen accumulation was higher in the controlled-release nitrogen blending treatments N135R2 and N135R3 than in the CK, indicating that the nitrogen release under the controlled-release nitrogen blending fast-acting nitrogen fertilizer treatment coincided better with the nitrogen requirement stage of rapeseed, and therefore it could absorb more nitrogen. In the two-year experiment, the apparent utilization of N fertilizer under N135R1-R3 treatments was higher than the other treatments. As one of the important indicators of nitrogen use by crops, N fertilizer utilization represents the extent to which N fertilizer contributes to yield. The results showed that controlled-release nitrogen fertilizer improved nitrogen utilization in rapeseed by promoting nitrogen uptake, while different controlled-release nitrogen blending ratios had different effects on nitrogen uptake and utilization. Soil microbial N is one of the key reservoirs of soil N and has an important role in regulating N turnover [27]. The soil microbial biomass N content under N135R2 and N135R3 treatments was significantly higher than that of the CK, indicating that the treatment with a 50–70% controlled release N blending ratio could effectively extend the N release cycle in the late stage of rapeseed fertility, which could continuously increase the soil microbial N reservoir capacity and reduce N loss [28]. Although soil microbial biomass carbon is a small proportion of the soil carbon pool, it is a large source of supply for soil nutrients [29]. It has been shown that soil microbial carbon is closely related to nutrient cycling of C, N, P, and S in soil [30]. Therefore, the controlled-release N blending treatment and the pure fast-acting N fertilizer treatment may affect soil nutrients differently. But more importantly, soil microbial biomass carbon was significantly higher under the N135R2 and N135R3 treatments than under the CK at the flowering stage. The flowering period is the overdue period from nutritional to reproductive growth of rapeseed, during which the plant grows vigorously and nutritional demand is high. Therefore, the higher soil microbial biomass carbon content under N135R2 and N135R3 treatments during the flowering stage was beneficial to improve soil nutrient availability and provide sufficient nutrients for rapeseed reproductive growth. Soil urease activity is one of the important indicators of soil nitrogen supply intensity [31]. Nitrogen fertilizer can significantly increase soil urease activity, and an appropriate N application can keep the balance of N input and output. That is, increasing the urease activity in the late reproductive stage of rapeseed catalyzes the hydrolysis of urea to NH3 and further its conversion to NH_4_^+^, thus reducing nitrogen loss due to the peak of ammonium nitrogen occurring in the early stage [32]. Although there was no significant difference in urease activity between the control and the mixed controlled-nitrogen release treatments at the seedling stage, it was slightly higher at the flowering stage, especially under the N135R3 treatment. This indicates that blended controlled-release nitrogen fertilizer can maintain high urease activity while reducing conventional nitrogen fertilizer usage, thereby reducing nitrogen fertilizer input. This may be related to the change in soil microbial community composition caused by a blended controlled-release treatment [32]. In addition, studies have reported that controlled-release fertilizer increases the ammonium nitrogen content in soil, thereby promoting the increase of urease activity [33]. The results of this study are consistent with this conclusion. Similarly, soil invertase enzyme activity plays an important role in increasing soil labile nutrients in relation to organic matter, nitrogen and phosphorus content, microbial activity, and soil respiration intensity [33]. The level of soil enzyme activity reflects, to a certain extent, the conversion capacity of soil nutrients [34]. Therefore, the controlled-release blending treatment can improve the soil’s fast-acting nutrients in rapeseed in the middle and late stages, ensure soil metabolic activity, and improve soil nutrient supply capacity during seed development. Soil ammonium and nitrate nitrogen are the main forms of nitrogen that can be directly absorbed and used by crops, and their levels are closely related to crop nutrient uptake [35]. Different N fertilizer application measures had significant effects on soil nitrate and ammonium N content. The application of nitrogen fertilizer can significantly increase soil nitrate and ammonium nitrogen content, but excessive application of nitrogen can lead to the accumulation of large amounts of nitrate and ammonium nitrogen in the soil of the tillage layer. If the crops do not absorb and utilize the accumulated nitrate and ammonium nitrogen in time, they will leach with precipitation layer by layer and cause groundwater pollution [36]. Numerous studies have shown that controlled-release nitrogen fertilizers have a positive effect on increasing soil ammonium and nitrate nitrogen content. In this study, soil ammonium and nitrate nitrogen contents were higher under N135R2 and N135R3 treatments, which helped to meet the nitrogen demand of rapeseed during its critical reproductive period [37]. This is mainly due to the fact that the controlled-release fertilizer can ensure a high ammonium nitrogen content in the soil for a longer period of time by blocking the direct contact between the inner core urea and the soil urease, thus slowing down the hydrolysis and ammonification process of urea [38]. The highest nitrate–N content at the seedling stage under the CK treatment and the lowest at the flowering stage proved that the nutrient release of fast-acting N fertilizer was mainly concentrated at the early stage of fertility. The nutrient supply capacity was not as good as N135R2 and N135R3 treatments at the later stage.

## 5. Conclusions

This study systematically evaluated the effects of different proportions of controlled-release nitrogen blended with conventional nitrogen fertilizer treatments on rapeseed yield, nitrogen fertilizer utilization efficiency, and soil carbon and nitrogen nutrients. An appropriate controlled-release nitrogen ratio can improve grain yield and fertilizer utilization efficiency, improving economic benefits and reducing environmental risks. In addition, the combination of controlled-release nitrogen and conventional nitrogen fertilizers increased soil microbial carbon and nitrogen content and invertase activity, indicating that an appropriate controlled-release nitrogen ratio improves soil nitrogen availability. Under the treatment of 50–70% controlled-release nitrogen blended with conventional nitrogen fertilizers, the grain yield and nitrogen absorption remained the same as those under conventional nitrogen fertilizer applications, and the nitrogen fertilizer utilization efficiency was significantly higher than the latter. Therefore, a 25% reduction can be achieved based on conventional nitrogen application in rapeseed. The application of controlled-release nitrogen blended with conventional nitrogen fertilizers with a mixing ratio of 50–70% can maintain the same grain yield level as conventional nitrogen application and improve agricultural ecological benefits. This is an effective way to reduce nitrogen fertilizer application and increase benefits in traditional rapeseed production.

## Figures and Tables

**Figure 1 plants-12-04105-f001:**
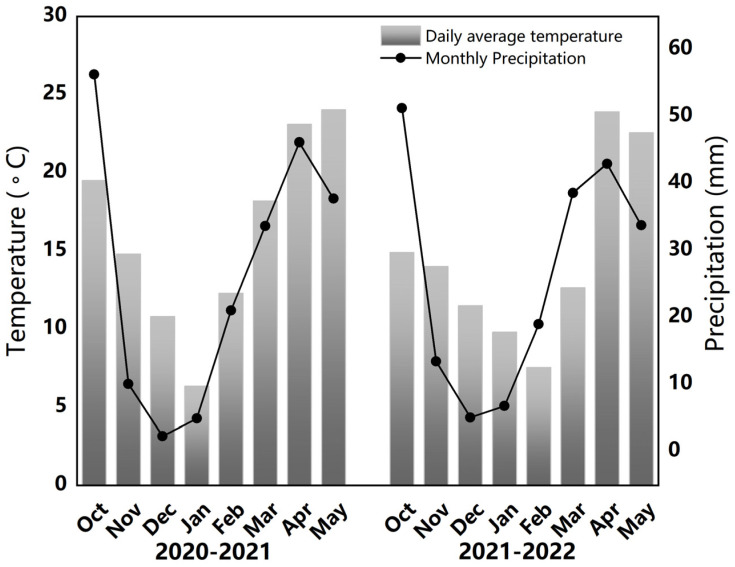
Monthly average temperature (°C) and precipitation (mm) over the two growing seasons.

**Figure 2 plants-12-04105-f002:**
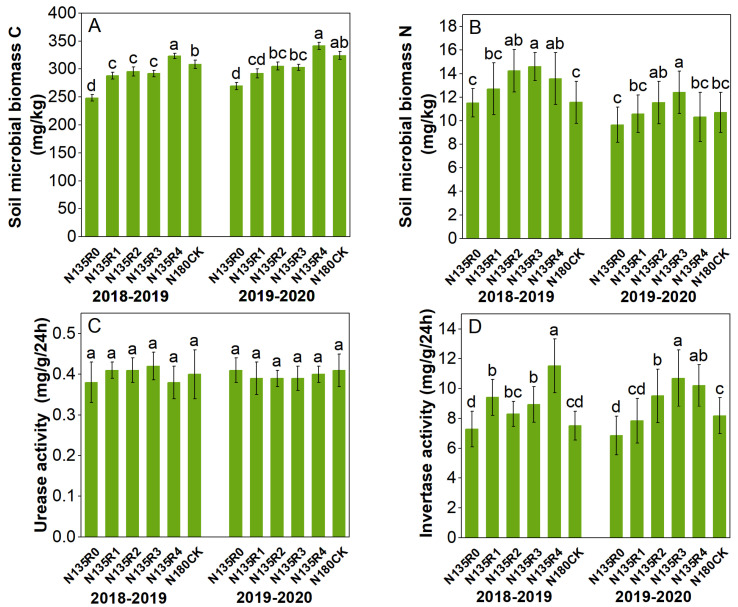
Effects of different fertilization treatments on soil microbial biomass C, N, and soil enzyme activity at seedling stage. N135: pure nitrogen (135 kg ha^−1^); R0: controlled-release nitrogen accounted for 0% of the total nitrogen application rate; R1: controlled-release accounted for 33% of the total nitrogen application rate; R2: controlled-release accounted for 50% of the total nitrogen application rate; R3: controlled-release accounted for 67% of total nitrogen application; R4: controlled-release accounted for 100% of the total nitrogen application rate; N180CK: pure nitrogen (180 kg ha^−1^). Different lowercase letters showed significant difference at *p* ≤ 0.05. (**A**) Soil microbial biomass carbon content, (**B**) Soil microbial biomass nitrogen content, (**C**) Urease activity, (**D**) Invertase activity.

**Figure 3 plants-12-04105-f003:**
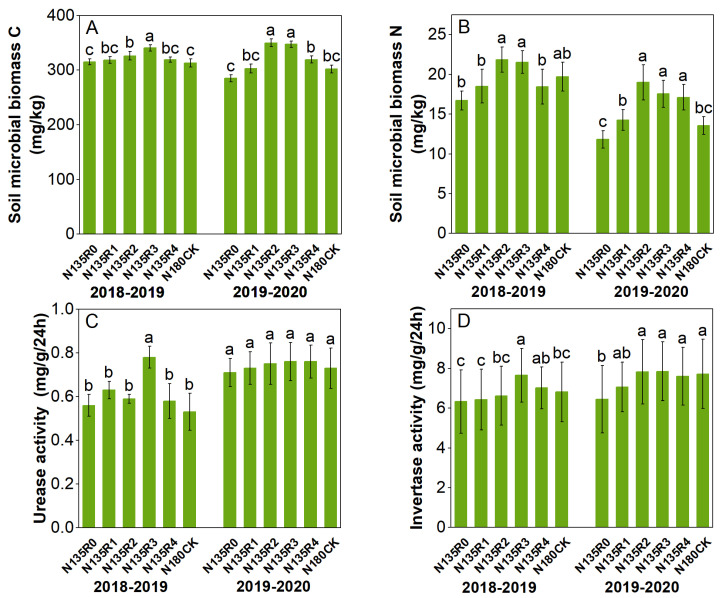
Effects of different fertilization treatments on soil microbial biomass C, N, and soil enzyme activity at flowering stage. Different lowercase letters showed significant difference at *p* ≤ 0.05. (**A**) Soil microbial biomass carbon content, (**B**) Soil microbial biomass nitrogen content, (**C**) Urease activity, (**D**) Invertase activity.

**Figure 4 plants-12-04105-f004:**
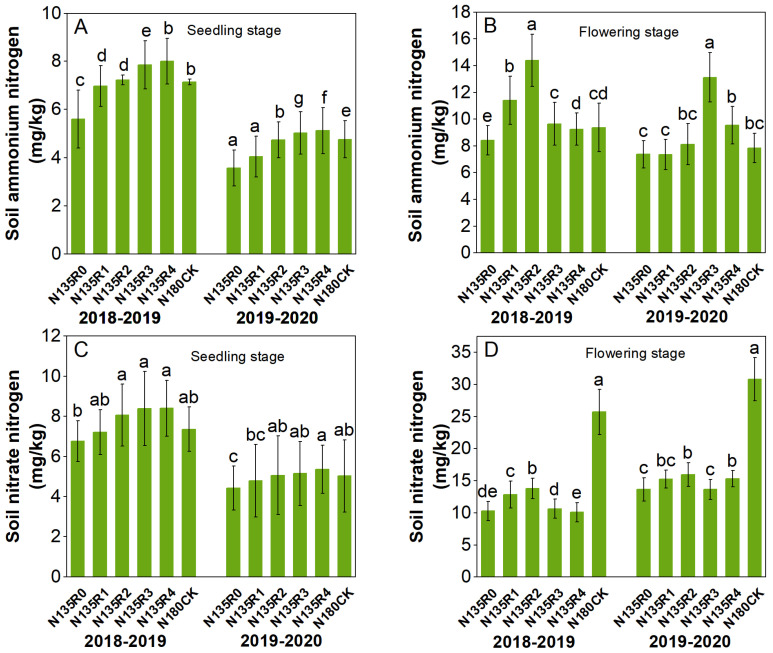
Effects of different treatments on soil inorganic N concentration. Different lowercase letters showed significant difference at *p* ≤ 0.05. (**A**,**B**) Soil ammonium nitrogen content, (**C**,**D**) Soil nitrate nitrogen content.

**Table 1 plants-12-04105-t001:** Effect of different fertilizer treatments on yield components of rapeseed.

Year	Treatment	Number of Pods Per Plant	Number of Seeds Per Pod	1000-Grain Weight (g)	Seed Yield (kg ha^−1^)	Biomass (kg ha ^−1^)
2018–2019	N135R0	170.67 c	19.8 b	4.07 a	2936.96 b	12,768.23 b
	N135R1	201.39 ab	20.53 ab	4.10 a	3404.11 a	13,816.25 ab
	N135R2	211.75 a	19.82 b	4.12 a	3635.91 a	14,146.75 a
	N135R3	205.83 ab	23.05 a	4.11 a	3447.18 a	13,490.25 ab
	N135R4	196.5 ab	21.58 ab	4.09 a	3027.54 b	12,869.17 b
	N180CK	208.42 ab	19.87 b	4.08 a	3492.47 a	13,449 ab
2019–2020	N135R0	161.83 c	21.11 b	4.09 a	3449.77 b	12,378.75 c
	N135R1	171.11 bc	21.49 b	4.12 a	3556.72 ab	13,458 ab
	N135R2	185.5 a	23.04 a	4.12 a	3859.11 a	13,796.5 a
	N135R3	173.83 b	22.61 a	4.09 a	3780.82 ab	12,626 bc
	N135R4	171.17 bc	21.73 b	4.11 a	3547.04 ab	12,180.25 c
	N180CK	182.44 ab	21.11 b	4.12 a	3822.93 a	12,520 bc

Note: Different lowercase letters showed significant difference at *p* ≤ 0.05.

**Table 2 plants-12-04105-t002:** Effect of different fertilizer treatments on nitrogen utilization in rapeseed.

Year	Treatment	Plant N (kg ha^−1^)	AEN (kg kg^−1^)	PFPN (kg kg^−1^)	NHI (%)	AREN (%)
	N135R0	162.33 c	4.88 c	19.4 b	68.76 b	49.11 d
	N135R1	185.33 b	7.71 a	25.59 a	76.96 a	78.58 a
2018–2019	N135R2	207.50 a	8.31 a	26.19 2 a	78.86 a	83.07 a
	N135R3	212.73 a	7.66 a	25.53 a	78.48 a	68.73 b
	N135R4	189.20 b	7.49 a	25.37 a	76.61 a	58.52 c
	N180CK	182.05 b	5.99 b	21.76 ab	75.04 ab	59.26 c
	N135R0	154.2 b	6.83 a	21.11 b	69.83 b	44.67 d
	N135R1	199.35 a	8.11 a	27.92 ab	73.32 ab	78.74 a
2019–2020	N135R2	212.52 a	9.2 a	28.73 a	76.86 a	74.71 a
	N135R3	201.73 a	7.62 a	26.35 ab	77.23 a	70.22 a
	N135R4	168.57 b	7.55 a	26.27 ab	73.99 ab	62.95 b
	N180CK	194.09 a	7.43 a	25.55 b	73.28 ab	56.35 c

Note: Different lowercase letters showed significant difference at *p* ≤ 0.05.

## Data Availability

Data are contained within the article.

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
