# Peer review of "Controlled-Release Nitrogen Mixed with Common Nitrogen Fertilizer Can Maintain High Yield of Rapeseed and Improve Nitrogen Utilization Efficiency"

_plants, 2023, doi:10.3390/plants12244105_

Round 1

Reviewer 1 Report

Comments and Suggestions for Authors

The authors of the manuscript showed the possibility of using slow-release fertilizers in the fertilization of rapeseed. Research in this area has been carried out for many years. 

I suggest that the authors of the paper formulate the research questions in the introduction and clearly define the objectives of the research. 

I suggest changing the phrase " normal fertilizers" to conventional fertilizers

Why was the measurement of mineral nitrogen limited to the topsoil only? 

Please expand on the characteristics of the slow-release fertilizer used. 

Conclusions should answer the formulated research questions.

Reviewer 2 Report

Comments and Suggestions for Authors

The authors of the manuscript by Hu and co-workers is mostly clear, easy to follow and, in its conclusions, justified. Here are some specific comments or questions:

1. Line 88, use the phrase"...can provide scientific support..."

2. Line 89: Is "light" used to mean "reduced", as in reduced fertilizer use or dosage?

3. Line 127: Can "in line with local practices" be specifically expanded upon? To any reader outside the "local" area, this is not at all informative.

4. Beginning with figure 2, the legend footnotes appear as a stand-alone paragraph separated from the one-line figure legend description. Following the one-line legend, you could write: Abbreviations: N135, ....." and include these descriptions in the same block of text as the one-line description. Also, the descriptions can be condensed because there's a lot of repetition of the same phrases. When these descriptions are identical, for example in Figures 1 and 2, Figure 2 could just direct the reader to the abbreviations already defined in Figure 1.

5. Section 3.2. This is really confusing. Fig. 4 has panels A, B, C and D, and these are not defined or referred to in the text or figure legend. I wasn't sure which panel the text descriptions were referring to.

6. Line 227: maximum yield of seed or "biological" (biomass).

7. Line 246: meaning of "nitrogen partial productivity"?

8. Table 2. Except for NHI, the abbreviations for the parameters in the table don't match those described in Methods, Section 2.4.3.

9.Line 269, insert a comma after (N135), continue the sentence with "while".

10. Discussion of urease, around lines 320. This seems to be a generally discussion of the significance of urease activity in soil, but is not related to the results actually obtained.

I find the results obtained are believable and their interpretation fair, with some need for clarification in certain parts as I outlined above. The Introduction did a good job in justifying the need for the work.

Comments on the Quality of English Language

A few phrases need to be clarified or changed, as I outlined in my comments to the Authors.

Author Response

请参阅附件

Round 2

Reviewer 2 Report

Comments and Suggestions for Authors

In the revised version of the manuscript, the authors have made the suggested changes.

Comments on the Quality of English Language

Essentially okay.